# Investigation of Track Gauge and Alignment Parameters of Ballasted Railway Tracks Based on Real Measurements Using Signal Processing Techniques

**Szabolcs Fischer** [1,*] , **Nándor Liegner** [2] , **Péter Bocz** [2] , **Ákos Vinkó** [2] **and György Terdik** [3]

1 Department of Transport Infrastructure and Water Resources Engineering, Faculty of Architecture, Civil and Transport Engineering, Széchenyi István University, 9026 Győr, Hungary

2 Department of Highway and Railway Engineering, Faculty of Civil Engineering, Budapest University of Technology and Economics, 1111 Budapest, Hungary

3 Department of Information Technology, Faculty of Informatics, University of Debrecen, 4031 Debrecen, Hungary

* Correspondence: fischersz@sze.hu; Tel.: +36-(96)-613-544

**Abstract:** This paper deals with the time-frequency characteristic analysis for track geometry irregularities using field data recorded by a comprehensive track inspection train. The parameters of the track gauge and the left and right rail alignment are considered to identify their characteristic wavelengths and the locations of their waveforms. In addition to the conventional time and frequency domain analysis, auto-adaptive signal decomposition techniques are used on four pre-selected track sections. During the time series analysis of the track gauge, the cumulative difference from the mean value is calculated, which makes it possible to distinguish the track section constructed with non-standard initial track gauges. The sensitive wavelengths of the track irregularities are obtained from the proper allocation of wavelength ranges in the Fourier Amplitude Spectrum of the original signal and the Fourier transform of the components detected by the Variational Mode Decomposition. This analysis can elucidate the wavelengths and positions of track irregularities that affect vehicle responses.

**Keywords:** railway track geometry; track gauge; alignment; signal processing; mode decomposition

## 1. Introduction

In railroads, electricity consumption [1–3] will be significant because of the worldwide energy crisis [4] and the expensive energy sources [5–8]. Therefore, it is a logical decision if railway companies want to save money by reducing the overconsumption of electricity. On the other hand, it is only one side of the economic railway operation (i.e., the vehicles). The other side is also a key factor: the railway infrastructure (permanent way, overhead wires, signaling, etc.). Of course, there is always a strong connection between all the parts [9–15]. It means that the worse the railway track geometry, the higher the needed traction energy demand of the vehicles, and so on. The current paper focuses on the railway from the above-mentioned areas in a special field.

An up-to-date knowledge of the technical condition of the railway track is essential to ensure the expected level of service and safety requirements and to optimize operating costs. Railway operators not only identify isolated track defects based on the limit exceedance of regularly measured track geometry but also schedule maintenance work by analyzing trends in measured time series data. In addition to the evaluation of local defects, they also calculate track quality indices over predefined lengths. In practice, fixed and moving window standard deviations are widely used [16,17], but some railway operators also use specific evaluation methods based on area under curve and space curve length calculations [18–20].

The traditional time series evaluation methods described above do not provide explicit information about the defects' wavelengths or shapes and their distance position on the railway line. Therefore, using frequency-domain (spectral analysis) and time-frequency-plane analysis methods is essential for analyzing the track geometry waveform. One of the main railway applications of spectral (frequency domain) analysis methods is the "de-coloring" of the track geometric parameters recorded by the versine (chord measuring) system and filtering them to a prescribed wavelength range(s) [17,21]. Another key application is the decomposition of the measured signal into wavelength components (modes) and the track position identification of each component. Decomposition methods include transformations with a predefined basis (FT, WT) and auto-adaptive methods (SVD, EMD, VMD). The meaning of the abbreviations and methods are described in the following paragraphs.

The Fourier Transform (FT) decomposes the signal into a linear combination of sine and cosine functions. The Power Spectral Density (PSD) produced by the Fourier transform can extract the dominant wavelengths and their amplitude from the track geometry parameters but does not provide information about their local representation. Due to the non-stationary nature of track geometry signals, the spatial/time dependence of the local wavelength information for FT can only be approximated by decomposing the measurement signal into near-stationary sections (STFT, GD—i.e., Gabor distribution) [22]. For STFT, however, the choice of the time window width is not straightforward, calibration is required to achieve the appropriate time and frequency resolution [23]. To be able to improve the estimation of PSD wavelength & amplitude characteristics, it is essential to remove outliers and trends (DC signal) in the track geometry data [24], which requires the application of procedures for the analysis of non-stationary signals.

The Wavelet Transform (WT) is an extension of the FT application to non-stationary and non-linear signals, where time-dependent frequency information of the signal is identified using the rescaled and modulated wavelet basis functions. Wavelet basis functions (mother wavelets) have different shapes and are localized in both the time and frequency domains. Therefore, only a limited choice of wavelets is available to meet the orthogonal decomposition and reconstruction—requirement of filter banks used in Discrete Wavelet Transform (DWT). In contrast, the Continuous Wavelet Transform (CWT) theoretically allows a considerable quantity of admissible wavelets without the orthogonality constraint. Using axle box acceleration data recorded on different track defects, Pablo et al. [23] compared the time-frequency resolution of MMW-based (Morlet's mother wavelet) based CWT power spectrum and the time-frequency resolution of the conventional STFT spectrum. The authors found that CWT provided a more detailed time-frequency distribution. Given the appropriate mother wavelet and scaling function choice, the DWT can effectively decompose the track geometry parameters into wavelength components as a dyadic filter bank. Using DWT-based signal decomposition, Zeng et al. [25] investigated the track profile irregularity of high-speed railways on a bridge structure. The characteristic wavelength and amplitudes of track misalignments were determined by the PSD of the components extracted from the decomposition using Daubechies' Mother Wavelet. Based on the above, the authors have identified the initial track geometry quality (manufacturing and construction imperfections), the irregularities at rail welds and insulation joints, and track adjustment errors of tamping machines when correcting track irregularities. Mariana et al. [26] investigated the mode decomposition of track geometry parameters (longitudinal level, alignment, track cant—or in other words: superelevation, and track gauge) using DWT by comparing different wavelet basis functions (daubechies, symlet, and coiflet). The authors found the Daubechies mother wavelet (DMW) to be the most suitable for characterizing the components of the track parameters mentioned above.

In contrast to the WT, the adaptive signal decomposition techniques do not require the proper selection of base functions. The singular value decomposition (SVD) is a data reduction tool in numerical linear algebra. In adaptive signal processing of time series, SVD is tracked from a recursively updated data matrix using a sliding window [27,28]. In this auto-adaptive decomposition algorithm, the basis is determined by the direction of

the most significant covariance present in the data. The resulting modes (components) are consequently ordered hierarchically, in order of decreasing importance for the system.

The Empirical Mode Decomposition (EMD) [29] with the Hilbert Transform (HT) is suitable for the multilevel decomposition analysis of non-linear and non-stationary measurement data. The sifting process of EMD is entirely data-driven, whereby the signal is separated into a finite set of intrinsic oscillatory modes. However, the method cannot separate frequency-modulated components, especially when both color and white noise are present in the signal. The Ensemble Empirical Mode Decomposition (EEMD) is an improvement of the basic method, which can handle signals with noise by statistically estimating the noise components [30]. Ling et al. [31] produced the signal components of the track geometry parameters using EEMD and then applied a Sliding Window based Adaptive SVD Algorithm to reduce the effect of mode mixing. Using this method, the authors could separate the non-linearly modulated frequency components. Another key issue with the EMD method is the criterion in the iterative sifting process to determine IMF components and the appropriate selection of the residual (trend) component. A widely used method for determining the trend in a signal by EMD is to calculate the pairwise cross-correlation of adjacent IMF components, followed by identifying the first pair of closely correlated components [32,33]. Liu et al. [34] applied several different time-frequency plane analysis methods to an artificially generated non-stationary signal with predefined components, but no qualitative comparison was given. The artificial signal was defined based on a maglev track's real unevenness measurement results.

Based on the above literature, this paper proposes deals with the time-frequency characteristic analysis for track geometry irregularities using field data recorded by a comprehensive track inspection train. The parameters of the track gauge and the left and right rail alignment are considered to identify their characteristic wavelengths and the locations of their waveforms. This additional information about the waveforms of rails can be beneficial in railway engineering practice to highlight locations with initial track stability problems and also support the deep investigation of the causes of track irregularities. This paper is organized as follows. First, the applied data processing techniques are introduced in Section 2, and in addition to the concept of the track degradation mechanism, the track stability issues are summarized in Section 3. Then, in Section 4, the steps to investigate track irregularities using the conventional and auto-adaptive signal processing techniques are compared and validated using field track irregularity data from the track inspection vehicle. Finally, Section 5 gives the main conclusions.

## 2. Signal Processing Techniques

### 2.1. Time/Spatial Domain Analysis

2.1.1. Cumulative Difference from the Mean Technique

The cumulative deviation from the mean is a mathematical method used to define homogeneous sections on a series of measurement data. The method can detect a section's typical deviation from the mean of the whole sample. The method summarizes the deviation of the individual measurement data from the mean. The formulae used are:

Consider a series of data $x_i$ with mean in Equation (1).

$$\overline{x} = \frac{\sum_{i=1}^{n} x_i}{n} \tag{1}$$

In Equation (1) $\overline{x}$ represents the mean, $n$ is the number of items in the data set of $x$, and $x_i$ means the $i$th element in the dataset of $x$.

The function of cumulative deviation from the mean will have as many elements as the original data set had. The elements are denoted by Equations (2) and (3).

$$S_1 = x_1 - \overline{x} \tag{2}$$

$$S_i = S_{(i-1)} + x_i - \overline{x}, \; if \; i > 1 \tag{3}$$

In Equations (2) and (3) $S_1$ is the deviation of the first sample value from the mean of data set $x$, while $S_i$ is the new series of points formed by adding up the differences consecutively.

As can be seen from the formulas, if the deviation from the mean is typically smaller in a section, the cumulative value is skewed to the negative; if the values are typically more extensive, the cumulative value is skewed to the positive. Plotting the values in a graph allows it to visualize the distinct sections within the data series. The slope of the graph also shows the magnitude of the trend deviation from the mean.

### 2.1.2. Space Curve Length-Base TQI Approach

The value of the rail alignment parameter of the track measurement graph shows the deviation from the straight line, measured separately on each rail. In the graph, the value on the baseline (value 0) indicates a perfect track. Error locations can also be identified by comparing the curve length of the graph with the length of the baseline (i.e., the error-free track) for a certain section. A more considerable quotient indicates a larger error.

In this paper, the method is used to investigate the deviation of the two rails' "rail direction", or in other words, the rail alignment parameter. First, the lengths of the measurement graphs of both rails are determined using the moving window method with multifold window lengths in the range of 3–25 m, and then the ratio of the curve length values obtained for the two rails is calculated. The quotient of the length of the two curves indicates if there is a different magnitude of error in the two rails. For example, Figure 1 shows the significant difference between the values measured on the two rails between 873+25 and 873+30 m. The ratio of the curve lengths of the two rails can also be indicated by a color scale, as shown in Figure 2. (The red color means better, hence the green color worse values).

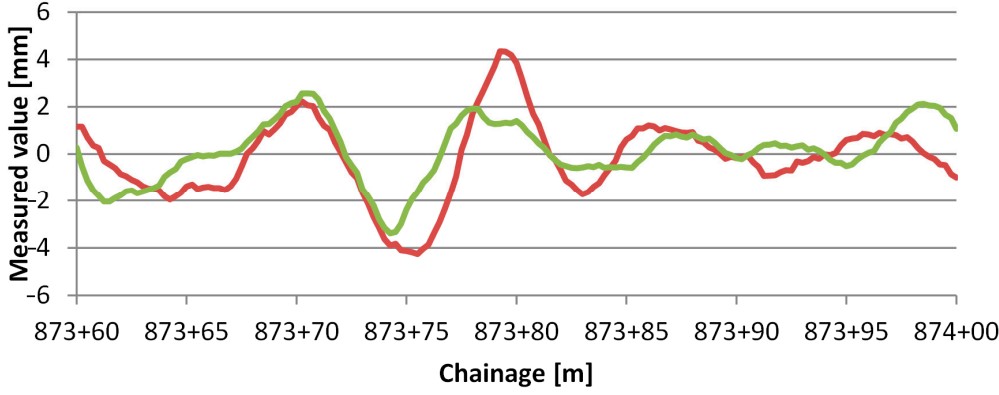

**Figure 1.** A subset of the rail alignment values measured on the two rails (the different colors mean the left and right rails, respectively).

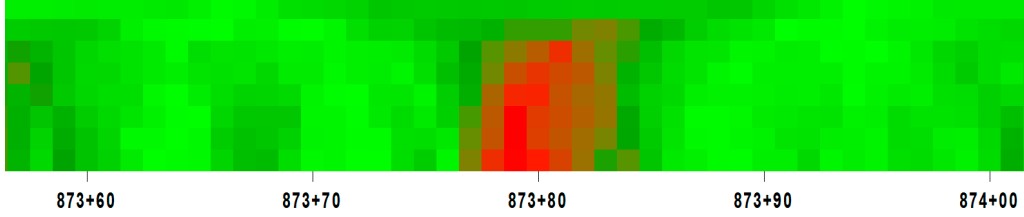

**Figure 2.** A subset of the rail alignment values measured on the two rails by color scale.

Discussing this topic may be interesting and important to mention the defects in laying railway tracks in connection with its broader perspective. It is mainly related to the horizontal (and partly vertical) structural stability of CWR (continuously welded rail) tracks [35]. By this, it is understood that the traditional calculation methods of analytical buckling and buckling stability (structural stability) calculations are based on assumed, theoretical, so-called sinusoidal (sine and sine transformed function) waveforms. It is

known from the international literature [36] that the world's first track buckling tests were carried out in Japan in 1932, and subsequently, the first stability calculation theory was based on them in 1957. It is worth noting that the theory of CWR tracks was developed in 1934, and a 4.2 km section of track was built on this basis in 1937, also in Japan. In 1953, CWR track began to be built in a standard way, resulting in nearly 8000 km of railway track being built by 1983 by the Japanese National Railways (JNR). Since 1953, CWR track has been used as a daily engineering practice in most countries because its operating costs are significantly lower than those of traditional fishplate-jointed track.

In Hungary, the theory of stability of CWR tracks was studied by E. Nemesdy in the 1960s [37]. In international practice, the theory of Meier [38], developed in 1937, and the modified Meier-method [39] are generally applied. In addition to these methods, there are several similar calculation methods [39], e.g., those of Engel [40], Raab [41], and Kerr [42]. These methodologies differ from each other only in a few initial values, and the results are also very close to those of the Meier-method.

The authors highlight only one example from the many international literature, Miura [36]. This paper contains a detailed list of the waveform of faults to be considered for track stability: from the first to the fourth waveform. On these, the local amplitude (local maximum, peak) values and the maximum and zero positions of the wave abscissas are given in an obvious way.

When considering track buckling of CWR tracks, as well as their structural stability, nowadays sophisticated methods can be used. Among others, FE (finite element) method [43], DE (discrete element) method [44], numerical methods [45,46], etc.

In this paper, the authors assume that measured signals similar to these waveforms of geometrical faults can be obtained, both in terms of track gauge and alignment track geometry parameters. However, this assumption is not the starting point for the signal processing analysis of the alignment and track gauge geometry parameters.

### 2.2. Adaptive Signal Decomposition Techniques

The EMD is a data-driven auto-adaptive method that decomposes signals into components referred to as Intrinsic Mode Functions (IMF) and a residual. IMFs are satisfying the following conditions:

- in the whole dataset, the number of extrema and the number of zero crossings must either equal or differ by at most one;
- at any point, the mean value of the envelope defined by the local maxima and the envelope defined by the local minima is zero.

The construction of IMFs is not unique. In the authors' current calculations, we apply the most popular version, which is implemented in MATLAB using the so-called sifting process method, see [29]. One of the outstanding advantages of EMD is that the modes (IMFs) are orthogonal. The sum of all modes and the remaining residual resemble the original signal. The IMFs fluctuated around zero. The only non-zero mean component is the remaining residual. An IMF represents a simple oscillatory mode. It has been noticed [47] that the frequency bandwidths tend to reduce as the number of IMFs increases.

Moreover, it is shown that Mutual Information between consecutive IMFs can be used for measuring the dependency between variables. Hence if the mutual information is low, then these two IMFs are stochastic. Besides the EMD method, the Variational Mode Decomposition (VMD) can be applied to describe the properties of a time series [48]. The VMD has a slightly more restrictive IMF than EMD and uses robust preprocessing for peak detection, then performs spectrum segmentation based on detected maxima. The modes of VMD have limited bandwidth.

The above decompositions of a nonlinear and nonstationary signal make possible the application of Hilbert Transform (HT) for the IMFs and, since the additivity, for the whole signal providing the Hilbert-Huang Transform (HHT). The HT maps the temporal-space data to time-frequency space, where both the amplitudes and frequencies depend on time as well. The marginal Hilbert spectrum can be computed from the Hilbert spectrum and

collects the total amplitude contribution to each frequency value. Although the frequency of the marginal Hilbert spectrum has a totally different meaning from the Fourier transform, one can apply it to separate the trend from the signal [33], for instance.

## 3. Mechanism of Track Gauge Degradation in Hungary (Track Gauge Variation)

On behalf of MÁV (Hungarian State Railways), research and development (R&D) was carried out on the topic of "Research and investigation of the causes of gauge narrowing with finite-element modeling, in normal tracks and turnouts, as well as in site and laboratory conditions" [49] subject at the Department of Highway and Railway Engineering of BUTE (Budapest University of Technology and Economics). The most important results of the research related to the topic of the publication are summarized below.

MÁV CRTI Ltd. (Hungarian State Railways Central Track Inspection Ltd., Budapest, Hungary) provided a local gauge narrowing fault dataset obtained from their measurements. The results were evaluated by categories of the railway line, superstructure system, and track curvature. Based on the list of local gauge narrowing faults, five Hungarian mainlines were subjected to a more detailed examination. The following conclusions have been drawn from the averages and their annual changes.

Gauge narrowing is typical for straight railway sections. However, it is not generated in curved tracks. The reason for this is that the lateral force occurring in curves generally causes rail wear, which compensates for a possible narrowing of the track, as well as gaps due to the dimensional tolerances of the rail fastenings, which act in the direction of the gauge widening.

In terms of gauge narrowing, the MÁV 48.5 rail system superstructure is the most affected. The explanation is that these superstructures are pretty old (approximately 50–55 years old), which structures are standard on the main lines, and the aging processes are already increasing. Therefore, in straight sections, the gauge narrows as time progresses. The most affected are the concrete sleepers originally constructed with wooden inserts marked "T", and "L", as well as for the MÁV 48 rail system, where the aging of the wooden inserts and the ingress of water into the dowels initiate a serious deterioration process. The straight track sections with the UIC54 (mainly 54E1) and UIC60 (60E1) rail systems are younger and did not or only slightly increase gauge narrowing over time. On-site measurements of the track were carried out in terms of track gauge and rail profile at fifteen places with gauge narrowing. Seventeen concrete sleepers were selected, taken out of the track, and delivered to the department's laboratory for a more detailed investigation. From the measurements of the on-site gauges, rail web, rail foot, and distances between the ribs and the inner side surfaces of the GEO baseplates (so-called K-type rail fastening system), it is concluded that the improper position (reduction in the distance) of the baseplate plays a significant role in the development of gauge narrowing. This statement was particularly true for the old MÁV 48 rail system superstructures (sleepers marked "L" originally with wooden inserts).

Based on the executed laboratory tests, the GEO baseplates of the sleepers marked "L" could be moved by hand at an amplitude of 5 to 7 mm. These sleepers were manufactured originally with wooden inserts for MÁV 48 rail system. They were approximately 50 years old at that time. Theoretically, the movement could have been only 1 mm due to the hole-bolt diameter difference in respect of geometrical tolerance; however, due to the thinning of the bolts due to corrosion and the slight expansion of the holes, more significant movements may occur.

Another but a smaller cause of gauge narrowing is the change in rail inclination. The effect of the increase in rail inclination appears mainly in the 60 kg/m rail superstructure with direct elastic rail fastening, where the railpad was compressed mostly asymmetrically, thus contributing to the development of gauge narrowing by 1–1 mm per rail. The change in the rail inclination causes a maximum narrowing of the gauge of 2 mm per rail, as the rail inclination changes from 1:20 to approx. 1:16. Based on more detailed measurements, it can be concluded that it is rare for the inclination of both rails to increase markedly. During the

production of track structural elements, accumulating dimensional tolerances of individual elements can have a detrimental effect on the dimensional tolerances of the final gauge. As a result of our tests, it is a recommendation that rail fastening systems consist of as few structural elements as possible on cross-sleeper tracks with ballast bed. Minimizing the number of elements can reduce the cumulation of dimensional tolerances and the degree of gauge narrowing.

During the mechanical FE (finite element) modeling of the rail, the lateral buckling of the rail was examined, resulting from the inhibited heat expansion in case the rail fastenings no longer hold down the rail base properly. An answer was tried for the question of what kind of deformation the rail may suffer in the track due to the failure of subsequent rail fastenings and how this contributes to the development of gauge narrowing. It can be concluded that the rail remains stable even for the maximum thermal force if the lateral support of up to a small, specific number of consecutive rail fastenings does not work properly. In the event of a more significant deficiency, the rail may become unstable, move sidewards and lose its stability.

A FE model was set up for concrete sleepers to examine how the effects occurring during the production of concrete sleepers, as well as after their installation in the track—during operation—affect the formation of gauge narrowing. The completion of the finite element modeling showed that the shrinkage of the cross-sleepers and other time-dependent shape changes do not contribute significantly to the development of the gauge narrowing. However, it was also examined how the uneven supports of the cross-sleepers, external support only, water bag, etc., affect the bent shape of the sleepers. As a result of this cumulative effect, significant narrowings can also occur. However, with careful construction and maintenance, high gauge narrowing values can be avoided by ensuring adequate and even support of the concrete sleepers.

As a result of performed investigations, it can be concluded that a gauge narrowing can be generated beyond the geometrical tolerances resulting from the unfavorable combination and accumulation of various minor effects.

## 4. Analysis of the Measurement Data and Discussion of the Results

During the analyses, real track geometry measurement data of a multi-functional track recording vehicle (TRV) were used. One of the busiest railway lines in Hungary was selected as a test section. TRV has a contactless, asymmetric chord-offset-based Track Geometry Measuring System (TGMS) synchronized with a Vehicle Dynamic Measuring System (VDMS) consisting of 24 accelerometers mounted on different structural parts of TRV [50]. The track geometry parameters measured by TRV meet the requirements of the EN 13,848 series. It can provide alignment and longitudinal level parameters as a raw versine measurement or as distortion-free data using the de-coloring procedure with additional band-pass-filtering to the D1 and D2 wavelength ranges. They can also be converted to any user-selected chord measuring system. Among the available track geometry parameters, only the track gauge (TG) and the alignment of the right (ALR) and left rails (ALL) were investigated in detail. The mean speed of the TRV during operation is 160 km/h. The sampling interval of each track irregularity data is 0.25 m along the track.

The following sub-sections of the railway line were selected for analysis, taking care to ensure that the sections were straight all the way along. The section numbers, length, and type of track superstructure are shown in Table 1. Part of Section 4 from 1659+20 has undergone a superstructure replacement.

**Table 1.** Investigated track sections.

| Section No. | Start Chain. * | End Chain. * | Length [m] | Speed ** [km/h] | Rail Type | Sleeper Type | Rail Fastening |
|---|---|---|---|---|---|---|---|
| 1 | 847+00 | 864+00 | 1700 | 160 | 60E1 | LW | Skl-1 |
| 2 | 873+00 | 884+00 | 1100 | 160 | 60E1 | LW | Skl-14 |
| 3 | 1054+00 | 1108+00 | 5400 | 160 | 60E1 | L2 | SKl-14 |
| 4 | 1620+00 | 1679+00 | 5900 | 160 | 54E1 | LW | Skl-3 |
| | | | | | 60E1 | L2 | Skl-14 |

* Hectometer chainages, i.e., 847+00 means 84.7 km. ** Operating Speed limit.

## 4.1. Designation of Homogenous Track Sections

The cumulative deviation from the mean was applied to the track gauge measured on the railway track and the "rail alignment" parameters measured on the two rails. The individual sub-sections were created using the gauge parameter. This method separated the track by nearly homogeneous sub-sections by gauge parameter. The calculations for the investigated four years (2017, 2018, 2019, 2020) are presented in Figures 3–6.

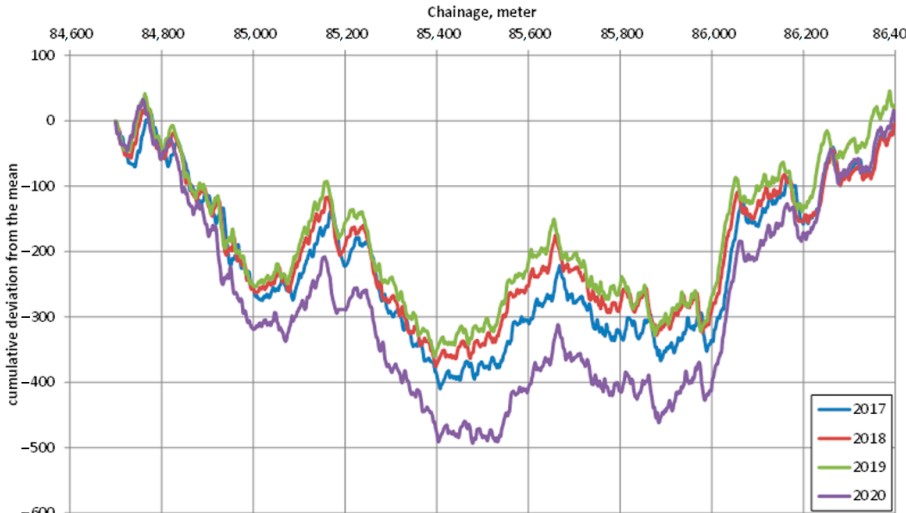

**Figure 3.** Cumulative deviation from the mean (section No. 1).

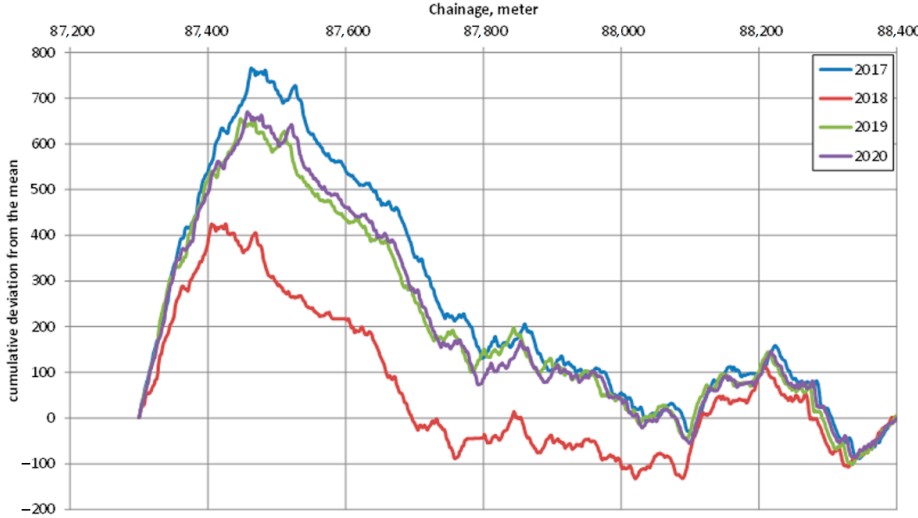

**Figure 4.** Cumulative deviation from the mean (section No. 2).

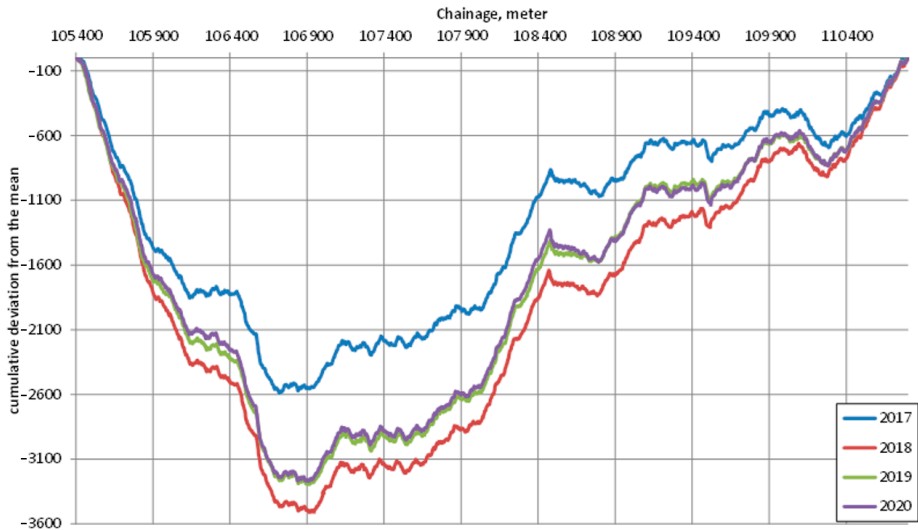

**Figure 5.** Cumulative deviation from the mean (section No. 3).

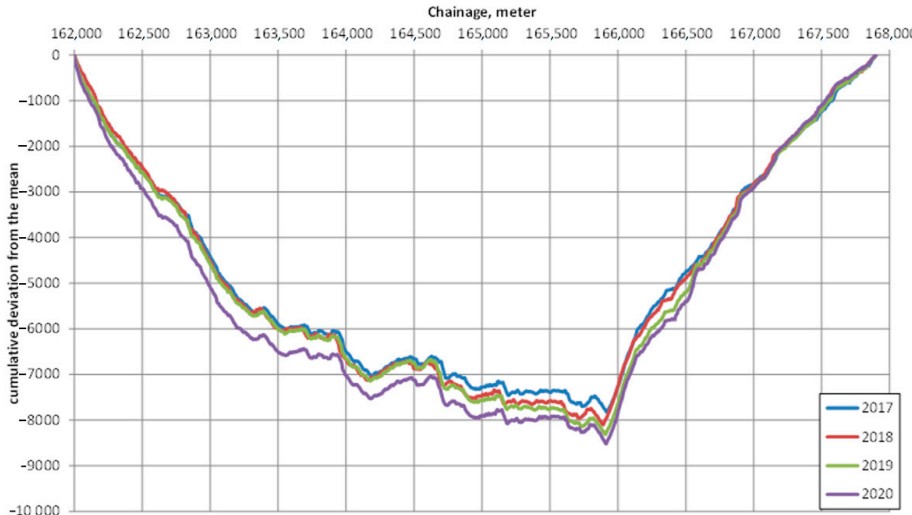

**Figure 6.** Cumulative deviation from the mean (section No. 4).

Special mention should be made of the measurement data series between chainage 1620+00–1679+00 because it is very clear that the superstructure was replaced from 1659+20 m onwards, and the gauge of the replaced sleepers was 1435 mm, compared to the 1433 mm gauge used by the MÁV in the past.

Based on a graph of the cumulative deviation from the mean, the sections with different gauge spacings were manually identified and delimited. The average gauge and rail alignment values were also calculated within the delimited sections. The values are shown in Tables 2–5.

**Table 2.** Average and deviation of track parameters (section No. 1).

| Start Chainage | End Chainage | TG | ALL | ALR | TG | ALL | ALR |
|---|---|---|---|---|---|---|---|
| | | Average Value [mm] | | | Standard Deviation [mm] | | |
| 847+00 | 854+00 | −0.64 | 0.76 | 0.99 | 0.74 | 0.58 | 0.71 |
| 854+00 | 860+00 | −0.47 | 0.83 | 1.05 | 0.72 | 0.60 | 0.80 |
| 860+00 | 864+00 | −0.32 | 0.95 | 1.02 | 0.74 | 0.66 | 0.80 |

**Table 3.** Average and deviation of track parameters (section No. 2).

| Start Chainage | End Chainage | TG | ALL | ALR | TG | ALL | ALR |
|---|---|---|---|---|---|---|---|
| | | Average Value [mm] | | | Standard Deviation [mm] | | |
| 873+00 | 874+80 | −0.50 | 1.02 | 0.97 | 1.08 | 0.84 | 0.73 |
| 874+80 | 878+00 | −1.35 | 0.74 | 0.66 | 0.73 | 0.61 | 0.49 |
| 878+00 | 884+00 | −0.97 | 0.80 | 0.82 | 0.80 | 0.67 | 0.63 |

**Table 4.** Average and deviation of track parameters (section No. 3).

| Start Chainage | End Chainage | TG | ALL | ALR | TG | ALL | ALR |
|---|---|---|---|---|---|---|---|
| | | Average Value [mm] | | | Standard Deviation [mm] | | |
| 1054+00 | 1061+40 | 0.89 | 0.69 | 0.79 | 0.69 | 0.51 | 0.65 |
| 1061+40 | 1064+45 | 1.56 | 0.68 | 0.80 | 0.65 | 0.48 | 0.57 |
| 1064+45 | 1067+15 | 0.83 | 0.91 | 1.03 | 0.88 | 0.84 | 0.83 |
| 1067+15 | 1084+80 | 1.94 | 0.74 | 0.92 | 0.72 | 0.61 | 0.70 |
| 1084+80 | 1102+75 | 1.80 | 0.78 | 0.94 | 0.69 | 0.66 | 0.74 |
| 1102+75 | 1108+00 | 2.13 | 0.59 | 0.91 | 0.66 | 0.48 | 0.63 |

**Table 5.** Average and deviation of track parameters (section No. 4).

| Start Chainage | End Chainage | TG | ALL | ALR | TG | ALL | ALR |
|---|---|---|---|---|---|---|---|
| | | Average Value [mm] | | | Standard Deviation [mm] | | |
| 1620+00 | 1633+20 | −1.83 | 1.05 | 0.92 | 0.96 | 0.78 | 0.70 |
| 1633+20 | 1659+20 | −0.98 | 1.11 | 1.08 | 1.08 | 0.90 | 0.81 |
| 1659+20 | 1679+00 | 0.24 | 1.10 | 1.14 | 0.98 | 0.90 | 0.91 |

The average gauge and rail alignment values for the sub-sections were also plotted in a graph (Figure 7), but no correlation was found between the two values. It means that the average value of the track gauge alone does not affect the magnitude of the rail alignment errors.

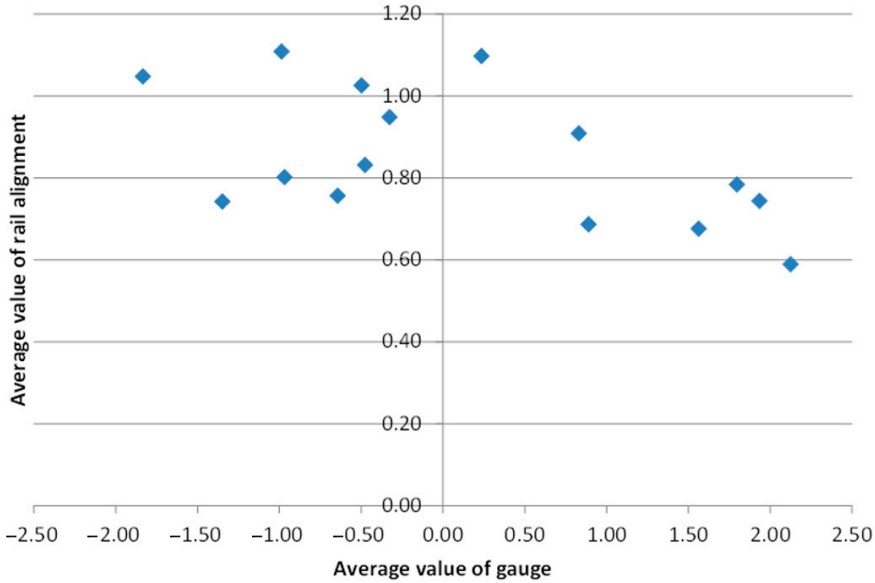

**Figure 7.** Average gauge and rail alignment values of sub-sections (all dimensions are in mm).

*4.2. Multi-Scale Decomposition Analysis*

4.2.1. Fourier Amplitude Spectrum (FAS)

The Fast Fourier Transformation (FFT) was applied to the pre-selected track sections (Table 1) for the frequency domain analysis of the investigated track geometric parameters. The raw data of the versine measuring system are used without "de-coloring" and filtering them. First, a Fourier Amplitude Spectrum (FAS) was generated for each track section and smoothed with the Konno-Ohmachi window (KOW) (see Figure 8), which is symmetric in log scale. The smoothing is performed by convolving the window function with the FAS [51].

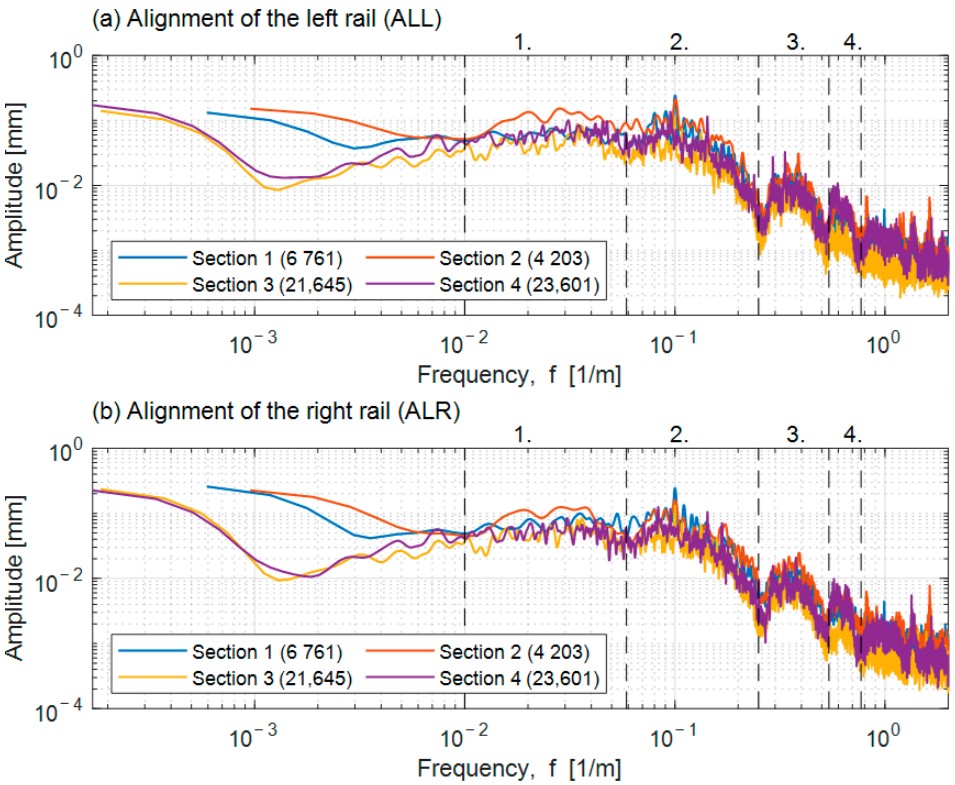

**Figure 8.** Fourier Amplitude Spectrum (FAS) of (**a**) ALL and (**b**) ALR on the investigated track sections (see Table 1) with the indication of the detected characteristic Wavelength ranges (modes) and the number of samples included in the investigated signal.

Based on the FAS, the inherent sensitive components and their characteristic wavelength range can be distinguished (Table 6) as the reciprocal of the frequency ($f$). The alignment components of the right rail (ALR) and left rail (ALL) parameters were determined by separate readings of each track section. The limits of the range ($\lambda_{min}$; $\lambda_{max}$) and the outliers in the range ($\lambda_{peak}$) are indicated for each extracted mode. The outliers are displayed in Table 6. in order of their weights, numbered sequentially. The wavelength ranges and the corresponding amplitudes of the ALL and ALR parameters belonging to the same track section show significant correlations. Comparing the wavelength range properties of these parameters related to the different track sections showed a similar close correlation. Still, the associated amplitude estimation accuracy differs significantly due to the different data numbers of each section. Therefore, all the investigated track sections highlight some discrete wavelength components.

**Table 6.** The identified Modes (wavelength ranges) of ALL and ALR based on their smoothed FAS.

| Mode No. | | Wavelength, λ [m] | | | | | | | |
|---|---|---|---|---|---|---|---|---|---|
| | | Section 1 | | Section 2 | | Section 3 | | Section 4 | |
| | | **ALL** | **ALR** | **ALL** | **ALR** | **ALL** | **ALR** | **ALL** | **ALR** |
| 1. | $\lambda_{max}$ | 83.33 | 76.92 | 96.15 | 87.72 | 90.91 | 84.75 | 100.00 | – |
| | $\lambda_{Peak1}$ | 24.39 | 27.78 | 30.00 | 33.90 | 28.57 | 28.57 | 30.30 | – |
| | $\lambda_{min}$ | 16.95 | 16.39 | 15.63 | 15.92 | 14.93 | 17.24 | 14.93 | – |
| 2. | $\lambda_{max}$ | 16.95 | 16.39 | 15.63 | 15.92 | 14.93 | 17.24 | 14.93 | 17.51 |
| | $\lambda_{Peak1}$ [1] | 10.00 | 10.00 | 10.00 | 10.00 | 10.53 | 10.00 | 10.53 | 10.00 |
| | $\lambda_{Peak2}$ | 9.09 | 9.10 | 8.92 | 8.90 | 9.34 | 9.35 | 6.99 | 7.00 |
| | $\lambda_{Peak3}$ | 8.40 | 8.33 | 8.12 | 8.20 | 8.11 | 7.52 | – | – |
| | $\lambda_{min}$ | 3.85 | 3.85 | 3.85 | 3.75 | 3.85 | 3.95 | 3.76 | 3.72 |
| 3. | $\lambda_{max}$ | 3.85 | 3.85 | 3.85 | 3.75 | 3.85 | 3.95 | 3.76 | 3.72 |
| | $\lambda_{Peak}$ [1] | 2.78 | 2.78 | 2.54 | 2.54 | 2.73 | 2.56 | 2.95 | 3.03 |
| | $\lambda_{min}$ | 1.85 | 1.92 | 1.92 | 1.92 | 1.92 | 1.89 | 1.85 | 1.88 |
| 4. | $\lambda_{max}$ | 1.85 | 1.92 | 1.92 | 1.92 | 1.92 | 1.89 | 1.85 | 1.88 |
| | $\lambda_{Peak}$ | 1.59 | 1.59 | 1.54 | 1.54 | 1.59 | 1.60 | 1.72 | 1.75 |
| | $\lambda_{min}$ | 1.30 | 1.35 | 1.30 | 1.30 | 1.30 | 1.26 | 1.33 | 1.37 |

[1] Dominant discrete wavelength (center (carrier) frequency).

The spectral peaks in mode number 1 of the alignment parameters are approximately at the upper limit of the D1 range specified by the standard (λ = 25–35 m) [16]. In mode number 2 of ALL and ALR, the discrete wavelength of λ = 10 m is dominated in all investigated track sections. In addition, some smaller weighted components can be identified in the interval of λ = 7.5–10.0 m. The peak values in mode number 3 converge around λ = 2.5 m. This value corresponds to the bogie wheelbase of the rolling stock on the line. The track gauge and alignment parameters in track Section 3 showed consistency in the wavelength range of mode number 4 (λ = 1.20–1.80 m). Considering the standard 60 cm sleeper spacing, this means 2–3 pieces of sleepers.

No similar narrow frequency bands were identified for the TG parameter. The FAS of the TG shows the harmonic overtones of sleepers' standard sub-spacing (60 cm) with monotonically increasing amplitude (Figure 9). In track section No. 4, the multiples of the nominal sub-spacings of sleepers are highlighted by the amplitude spectrum with a tolerable accuracy (Table 7). In Figure 9, the spectral peaks that closely match the multiples of the sleepers' sub-spacing are marked with the corresponding serial numbers. The second part of track section no. 4 has sleepers accommodating the rails 2 mm narrower than the standard value (see Section 4.1). This baseline change in the TG parameter can cause the monotonic baseline variation of the FAS of TG.

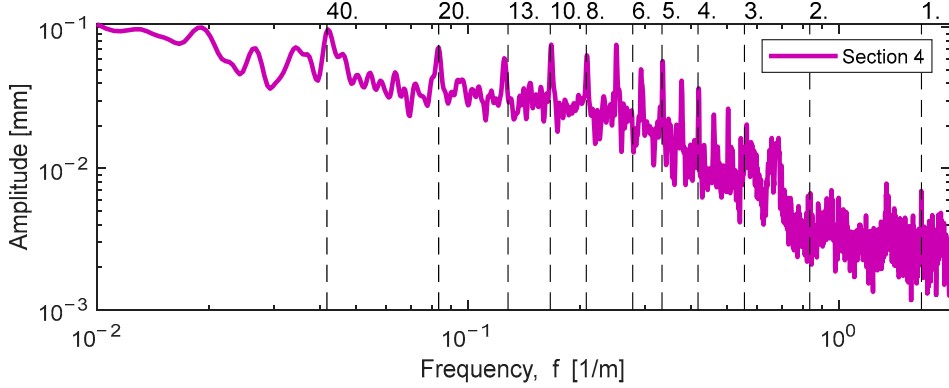

**Figure 9.** FAS of TG with the indication of harmonic oscillation modes of sleeper spacing distance.

**Table 7.** Discrete Wavelengths associated with Outliers in FAS of TG in Section 4 (see Figure 9).

| The Characteristic Discrete Wavelength of the Track Gauge (In Track Section 4) | | | | | | | | | | |
|---|---|---|---|---|---|---|---|---|---|---|
| λ [m] | 0.60 | 1.19 | 1.84 | 2.38 | 2.98 | 3.41 | 4.78 | 5.96 | 7.95 | 12.05 | 24.15 |
| n × k [–] | 1.0 | 2.0 | 3.0 | 4.0 | 5.0 | 6.0 | 8.0 | 10.0 | 13.0 | 20.0 | 40.0 |

4.2.2. Empirical Mode Decomposition (EMD)

The spectral properties of track gauge and alignment parameters introduce difficulties in applying the EMD approach. The alignment is a narrow-band signal with, depending on the track quality, several partially or fully interlaced wave components. In contrast, the track gauge has a monotonically increasing amplitude characteristic with increasing wavelength and includes additional harmonic peaks referred to as sinusoidal waveforms.

Stochastically occurring local defects in the railway track will intensify as the degradation process progresses, and more dominant wave components may appear or become detuned due to the track-vehicle interaction. The characteristic wavelengths of the track local irregularities can vary in a wide range, but they differ significantly from the general characteristics of the connecting sections in good quality. The local defects can be isolated point-like defects with small wavelengths or geometric irregularities with more extensive wavelength ranges. All elements of the track-vehicle system play a significant role in the development of the track irregularities. The above leads to the conclusion that the track geometric parameters are non-stationary.

The IMF components extracted by EMD cannot isolate the complex waveforms that appear at local defects. As a result, the adjacent wavelength ranges forming the local track irregularities are concentrated in one IMF component. The pairwise correlation variability of the IMF components also significantly illustrates this effect. During the shifting process of EMD, which is applied to track geometry parameters, the narrow-band signal characteristic is typically generated only for the second IMF component. The data series detail in Figure 10 highlights the first IMF component of the ALL in the fourth track section. This figure illustrates the significantly different wavelength ranges inherent in the mode. The amplitude spectrum of this IMF's FAS is shown in Figure 11a, while the sub-components are shown in Figure 11b–f. The subcomponents were determined using the Variational Mode Decomposition (VMD) [48] procedure. In the case of the TG parameter, the first IMF shows significant nonstationary features since all frequencies have nearly the same weight except for a short interval (Figure 12). Considering the above, it is clear that the IMF components extracted by the EMD shifting process require further data processing.

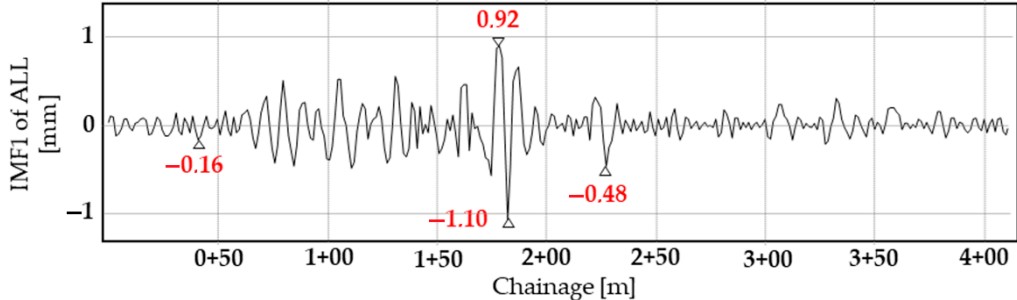

**Figure 10.** Sample section of EMD generated first IMF of ALL with short-wave and medium-wave irregularities of different extents, inherent mode mixing (track section No. 4).

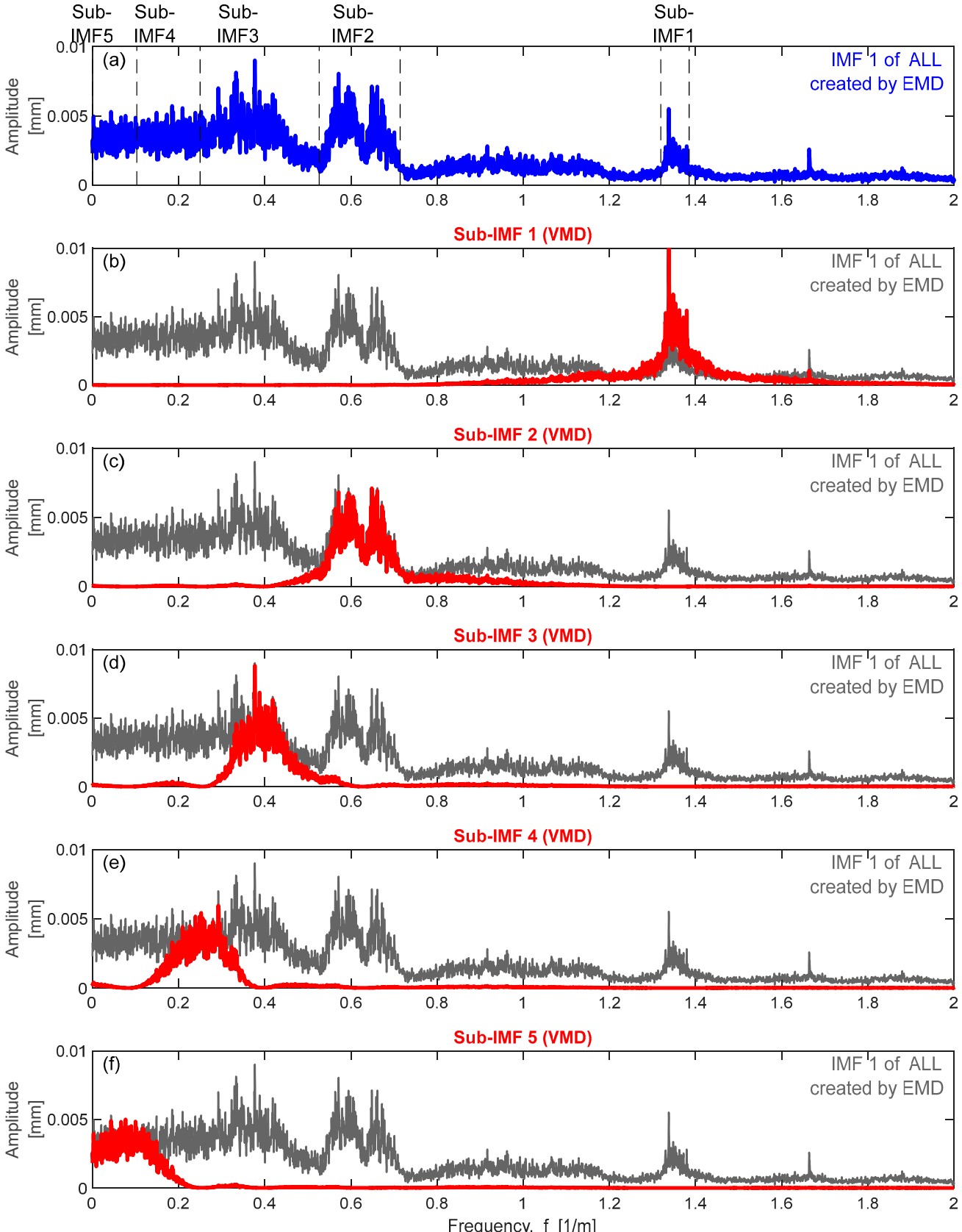

**Figure 11.** VMD's spectral decomposition of the first IMF extracted from ALL using EMD indicates the inherent sub-IMF components (track section No. 4). (**a**) IMF of ALL created by EMD; (**b**) Sub-IMF 1; (**c**) Sub-IMF 2; (**d**) Sub-IMF 3; (**e**) Sub-IMF 4 and (**f**) Sub-IMF 5.

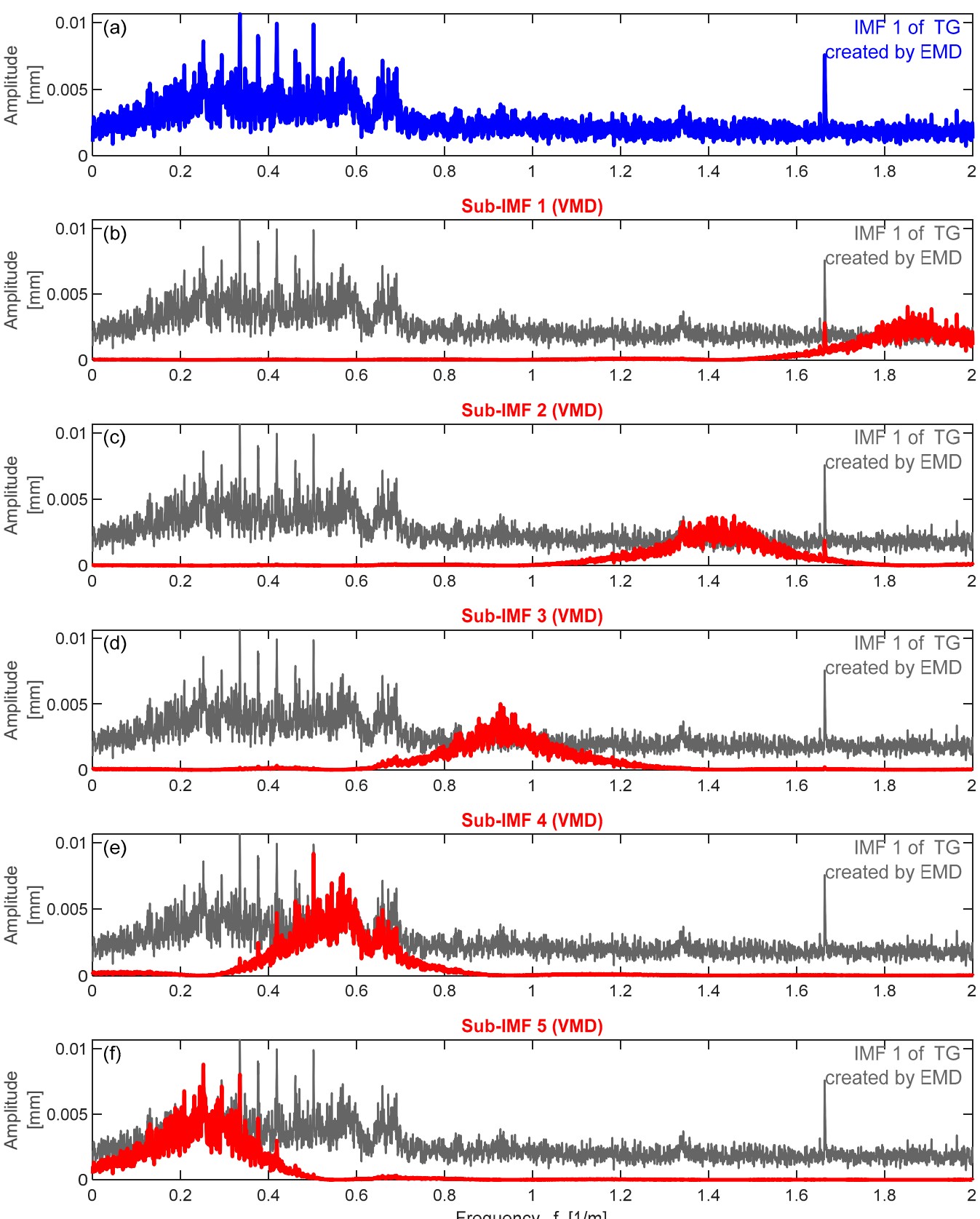

**Figure 12.** VMD's spectral decomposition of the first IMF extracted from TG using EMD indicates the inherent sub-IMF components (track section No. 4). (**a**) IMF 1 of TG created by EMD; (**b**) Sub-IMF 1; (**c**) Sub-IMF 2; (**d**) Sub-IMF 3; (**e**) Sub-IMF 4 and (**f**) Sub-IMF 5.4.2.3. Variational Mode Decomposition (VMD).

The ALL, ALR, and TG parameters are decomposed into components for all the four investigated track sections using VMD. For each IMF, the FAS spectra of the IMFs were determined as detailed in Section 4.2.1, and the ranges of the characteristic wavelength and the spectral peaks within it were summarized in Table 8. For all four main sections, it is clear that the difference between the wavelength ranges of the ALL and ALR parameters is negligible. However, the amplitude characteristics for a given wavelength may differ for the right and left rails. The related analysis is described in Section 2.1.2. The decomposition of the TG and ALL parameters are illustrated in Figures 13 and 14. It is also highlighted that the abrupt change in the baseline of the track gauge can be accurately extracted using VMD.

**Table 8.** Wavelength ranges in the IMF components of ALL, ALR, and TG calculated by VMD.

| Mode No. | | Wavelength, $\lambda$ [m] | | | | | | | | | | | |
|---|---|---|---|---|---|---|---|---|---|---|---|---|---|
| | | Section 1 | | | Section 2 | | | Section 3 | | | Section 4 | | |
| | | **ALL** | **ALR** | **TG** | **ALL** | **ALR** | **TG** | **ALL** | **ALR** | **TG** | **ALL** | **ALR** | **TG** |
| IMF0 | $\lambda_{max}$ | – | – | 1.38 | – | – | 0.75 | – | – | 1.09 | – | – | 0.76 |
| | $\lambda_{Peak1}$ | – | – | 1.01 | – | – | 0.61 | – | – | 0.77 | – | – | 0.60 |
| | $\lambda_{min}$ | – | – | 0.67 | – | – | 0.50 | – | – | 0.58 | – | – | 0.50 |
| IMF1 | $\lambda_{max}$ | 2.31 | – | 3.27 | | 2.34 | – | – | – | 2.40 | 1.89 | 1.87 | 1.99 |
| | $\lambda_{Peak1}$ | 1.88 | – | 1.81 | 1.81 | 1.58 | – | – | – | | 1.67 | 1.69 | 1.77 |
| | $\lambda_{Peak2}$ | 1.42 | – | 1.67 | 1.23 | 1.23 | – | – | – | 1.14 | 1.52 | 1.52 | 1.49 |
| | $\lambda_{min}$ | 0.89 | – | 1.05 | 1.01 | 1.03 | – | – | – | 0.82 | 1.36 | 1.38 | 1.19 |
| IMF2 | $\lambda_{max}$ | 3.63 | 3.69 | – | 3.62 | 3.75 | 3.93 | 3.90 | 3.80 | – | 3.69 | 3.66 | 3,67 |
| | $\lambda_{Peak1}$ | – | – | – | – | – | – | – | – | – | 3.42 | 3.42 | 3.42 |
| | $\lambda_{Peak2}$ | – | 2.90 | – | 2.85 | – | - | 2.85 | 2.94 | – | 2.99 | 2.99 | 2.99 |
| | $\lambda_{Peak3}$[1] | 2.57 | 2.66 | – | 2.57 | 2.55 | 2.32 | 2.55 | 2.56 | – | 2.39 | 2.65 | 2.66 |
| | $\lambda_{min}$ | 1.99 | 1.91 | – | 1.98 | 1.78 | 1.22 | 2.00 | 1.98 | – | 2.15 | 2.19 | 2.17 |
| IMF0 | $\lambda_{max}$ | – | – | 8.16 | – | – | 7.89 | – | – | 7.76 | – | – | 6.93 |
| | $\lambda_{Peak1}$ | – | – | 5.06 | – | – | 4.93 | – | – | 4.38 | – | – | 4.77 |
| | $\lambda_{min}$ | – | – | 2.14 | – | – | 2.57 | – | – | 1.93 | – | – | 2.17 |
| IMF3 | $\lambda_{max}$ | 9.13 | 8.67 | – | 10.10 | 10.40 | – | 9.25 | 9.41 | – | 9.16 | 10.07 | – |
| | $\lambda_{Peak1}$ | – | – | – | – | – | – | – | – | – | 7.01 | 6.99 | – |
| | $\lambda_{Peak1}$ | 5.87 | 7.10 | – | 7.15 | 6.77 | – | 7.52 | 7.52 | – | 6.02 | 5.96 | – |
| | $\lambda_{min}$ | 4.18 | 4.01 | – | 4.04 | 3.96 | – | 4.93 | 4.73 | – | 5.11 | 5.01 | – |
| IMF4 | $\lambda_{max}$ | 35.20 | 35.79 | 41.22 | 30.01 | – | 55.29 | 18.66 | 19.05 | 30.30 | 38.56 | 39.26 | 35.97 |
| | $\lambda_{Peak1}$ | – | – | – | – | – | – | – | – | – | – | 12.02 | 12.05 |
| | $\lambda_{Peak2}$[1] | 10.00 | 10.00 | 8.94 | 9.91 | – | 10.01 | 10.00 | 10.06 | 10.35 | 10.52 | 10.49 | 10.38 |
| | $\lambda_{min}$ | 7.22 | 7.22 | 4.83 | 7.39 | – | 5.30 | 8.02 | 7.39 | 4.37 | 7.52 | 7.48 | 4.78 |
| IMF5 | $\lambda_{max}$ | – | – | – | – | – | – | 90.18 | 117.63 | – | 88.49 | 131.11 | 86.76 |
| | $\lambda_{Peak1}$ | – | – | – | – | – | – | 27.19 | 28.93 | – | 32.59 | 37.58 | – |
| | $\lambda_{min}$ | – | – | – | – | – | – | 11.81 | 11.34 | – | 20.92 | 20.92 | 9.88 |

[1] Dominant discrete wavelength (center (carrier) frequency).

The wavelength range between 1.2–1.8 m in the ALL and ALR parameters is observed in all investigated track sections. In contrast, it occurs only in track section No. 4 for the track gauge parameter. This range of wavelengths is related to the standard sleeper spacing and its multiples. Therefore, the co-occurrence of this component in the alignment and track gauge parameter may indicate the presence of local defects.

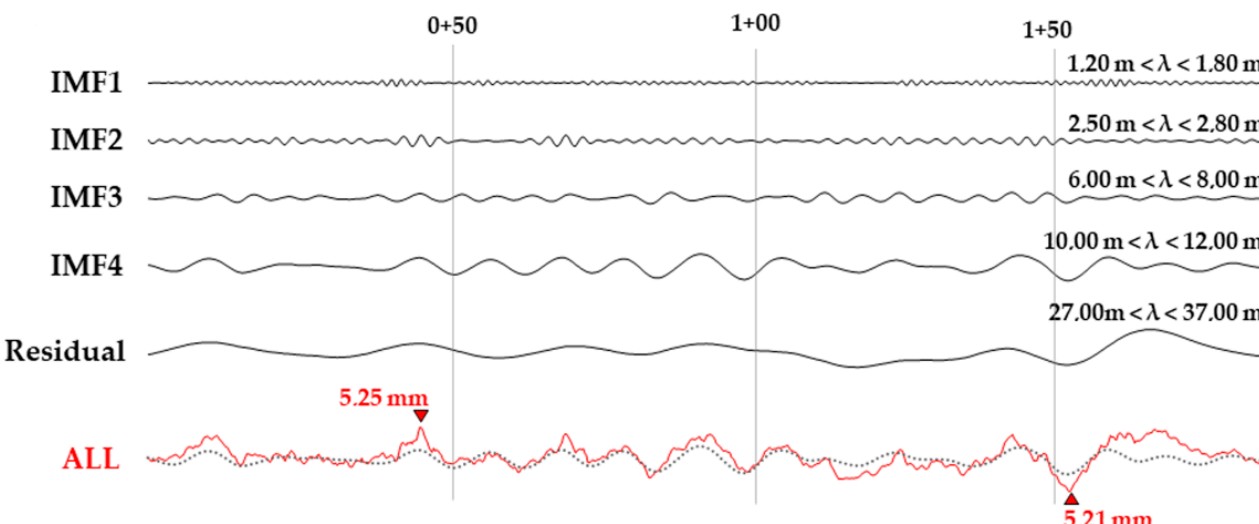

**Figure 13.** Illustration of the IMF components of ALL calculated by VMD. (X-axis represents distance along the track in [m] unit, while Y-axis represents ALL in [mm] unit).

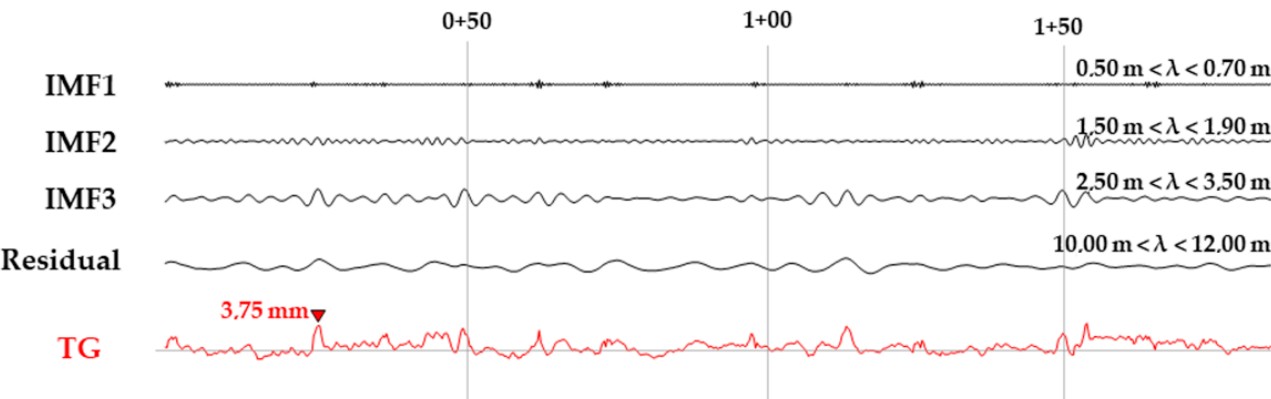

**Figure 14.** Illustration of the IMF components of TG calculated by VMD (X-axis represents distance along the track in [m] unit, while Y-axis represents ALL in [mm] unit).

The wavelength range of 2.5–2.8 m is present in all track sections' ALL and ALR parameters and in the TG parameter of main Section 4. This wavelength range is presumably related to the wheelbase of the vehicles on the line. For both alignment and track gauge errors, the design and dimensions of the bogie passing over them also have a significant excitation effect, as they amplify the smallest geometric deviation present in the track.

The wavelength ranges between 6.0 and 8.0 m and between 27.0 and 37.0 m were only detected in the alignment parameter. It is assumed that the track-vehicle interaction of the investigated sections is the cause of this phenomenon. Further investigation is needed to clarify the reasons.

The wavelength range between 10.0 and 10.5 m is prominent in the alignment parameter in all the track sections studied. This value is related to the Plasser 09-3X track tamping machine's chord offset layout used on the line. The control string length is 10.455 m. This assumption is further confirmed by the fact that this wavelength is also reflected in the modes of the track gauge. It is also important to highlight that this component was clearly identifiable in the FAS spectrum of the original signal.

## 5. Conclusions

Railway track irregularities can have complex waveforms amplified by the self-excited degradation process of the vehicle-track coupled system. Therefore, time and frequency domain analysis methods were used to analyze the waveforms of these defects. In our

investigations, particular attention was paid to determining the sensitive wavelength of local track defects related to the alignment and track gauge parameters, performed on four straight track sections and their subsections using real measurement data.

During the time series analysis of the track gauge, the cumulative difference from the mean value is calculated, which makes it possible to distinguish the track section constructed with non-standard initial track gauges. TQI can be used to identify track defects and can also be used to distinguish between right and left rail waveforms.

The alignment parameter is a narrow-band signal, while the track gauge has a broad-band frequency spectrum. The "manual" component separation and auto-adaptive component decomposition procedures based on Fourier Amplitude Spectra of the above two track geometric parameters were investigated separately. The characteristic wavelength range of the components determined by the auto-adaptive methods was obtained using FFT-based smoothed FAS. Due to the nature of the shifting process, empirical Mode Decomposition cannot separate the intertwined wavelength components typical of local track defects. For this reason, several different wavelength ranges are typically stacked up in the first IMF component. In the case of the alignment parameter, several clusters and a few distinct small wavelength components appear in the first IMFs generated by EMD. The first IMF of the track gauge parameter generated by EMD showed significant non-stationary features.

Using Variational Mode Decomposition (VMD) on all track sections, the clustered components within the IMF extracted by EMD and the physically meaningful trend of the parameters could be separated, yielding comparable results to manual component separation based on FAS. Both the manual and the applied auto-adaptive method (VMD) clearly highlighted the following discrete wavelength components:

The average of the peak values between $\lambda$ = 2.3–2.7 m corresponds to the wheelbase of the vehicles on the line (2.5 m). The identified wavelength range 1.0 m < $\lambda$ < 3.0 m is the typical wavelength of track gauge and alignment errors in the surroundings of the rail joints.

The value $\lambda$ = 10.0 m is assumed to be related to the chord offset layout of the track-tamping machine in service on the line (Plasser & Theurer 09-3X b = 10.455 m).

In addition to the above, the identified spectral peaks in the extracted modes are assumed to arise from the mutual excitation of the track-vehicle system. Therefore, their investigation requires vehicle dynamics simulations.

This additional information about the waveforms of rail irregularities can be beneficial in railway engineering practice to highlight the locations with initial track stability problems and also support the deep investigation of the causes of track irregularities. Furthermore, this approach can also be applied in the future to extract the sensitive wavelength of manufacturing and construction imperfections and to characterize the frequent shapes of track irregularities formed on a tangent or curved tracks.

**Author Contributions:** Conceptualization, S.F., N.L., P.B., Á.V. and G.T.; methodology, S.F., N.L., P.B., Á.V. and G.T.; software, S.F., N.L., P.B., Á.V. and G.T.; validation, S.F., N.L., P.B., Á.V. and G.T.; formal analysis, S.F., N.L., P.B., Á.V. and G.T.; investigation, S.F., N.L., P.B., Á.V. and G.T.; resources, S.F., N.L., P.B., Á.V. and G.T.; data curation, S.F., N.L., P.B., Á.V. and G.T.; writing—original draft preparation, S.F., N.L., P.B., Á.V. and G.T.; writing—review and editing S.F., N.L., P.B., Á.V. and G.T.; visualization, S.F., N.L., P.B., Á.V. and G.T.; supervision, S.F., N.L., P.B., Á.V. and G.T.; project administration, S.F., N.L., P.B., Á.V. and G.T.; funding acquisition, S.F., N.L., P.B., Á.V. and G.T. All authors have read and agreed to the published version of the manuscript.

**Funding:** This research received no external funding.

**Data Availability Statement:** Not applicable.

**Acknowledgments:** The authors would like to thank MÁV Ltd. and MÁV CRTI Ltd. for their help with providing measuring data. This paper was prepared by the cooperation of research teams "BME-RAIL" and "SZE-RAIL".

**Conflicts of Interest:** The authors declare no conflict of interest.

**Abbreviations**

| | |
|---|---|
| ALL | alignment of the left rail |
| ALR | alignment of the right rail |
| BUTE | Budapest University of Technology and Economics |
| CRTI | Central Track Inspection |
| CWR | continuously welded rail |
| CWT | continuous wavelet transform |
| DC | direct current |
| DE | discrete element |
| DEM | discrete element method |
| DMW | Daubechies mother wavelet |
| DWT | discrete wavelet transform |
| EEMD | ensemble empirical mode decomposition |
| EMD | empirical mode decomposition |
| EN | European Norm |
| FAS | Fourier amplitude spectrum |
| FE | finite element |
| FEM | finite element method |
| FFT | fast Fourier transform(ation) |
| FT | Fourier transform |
| GD | Gabor distribution |
| HHT | Hilbert-Huang transform |
| HT | Hilbert transform |
| IMF | intrinsic mode function |
| KOW | Konno-Ohmachi window |
| MÁV | Hungarian State Railways |
| MMW | Morlet's mother wavelet |
| PSD | power spectral density |
| STFT | short-time Fourier transform |
| SVD | singular value decomposition |
| TG | track gauge |
| TGMS | track geometry measuring system |
| TQI | track quality index |
| TRV | track recording vehicle |
| VDMS | vehicle dynamic measuring system |
| VMD | variational mode decomposition |
| WT | wavelet transform |

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
