# Peer review of "Investigation of Track Gauge and Alignment Parameters of Ballasted Railway Tracks Based on Real Measurements Using Signal Processing Techniques"

_infrastructures, doi:10.3390/infrastructures8020026_

Round 1
Reviewer 1 Report
The paper deals with using signal processing techniques to analyze railway track gauges and alignment. The paper is well-written and requires only a few changes (e.g., in line 107, "he" should be "they"; there is a comma in section 2.1 title). The paper is generally interesting, but its structure needs to be revised since some elements of scientific documents need to be included or described.
The authors need to state the novelty of the paper in the introduction.
Section 2 is called methodology, but it only explains the techniques the authors used in their research in a literature review way.
Section 3 is also a literature review. The authors mention a graph but do not present it (lines 233-236).
Section 4 is methodology.
The authors do not present conclusions at the end of the paper. They must revise it, showing their contribution to the field and recommendations for further work.
The authors should reorganize the paper and present a paragraph explaining its structure at the end of the introduction section (after the study's objective).
Author Response
See the attached PDF file.

Reviewer 2 Report
Dear authors,
it was a pleasure to read your work. In fact, it not only presents scientifc soundness as well as i believe it could add to the thematic literature.
However, I recommend the authors to add/emphatize the research limitations and future study lines.
best,
Author Response
See the attached PDF file.

Reviewer 3 Report
The introduction stresses the impact of track geometry energy on energy consumption. However its influence on traffic safety seems to be more important.
Please check the numbering of figures, references to them and the figure titles. For example in page 4 (line 149) the Fig. 2 is quoted instead of Fig. 1. Similarly please check the quotation in line 152.
It is suggested to describe the source of data used in the analysis (what kind of measuring car) and the tools used for on-site measurements nad their scope.
Lack of conclusions - for example on usability of the proposed methods in day-to-day railway track maintenance practice.
Check the numbering of chapter 6 (it should be number 5).
In references it is suggested to mention the fundamental works of ERRI D202 committee "Improved knowledge of CWR, including switches" and works of Kish, Samavedam and Esveld on this subject
Author Response
See the attached PDF file.
